# Essential role of accessory subunit LYRM6 in the mechanism of mitochondrial complex I

Etienne Galemou Yoga[1,2], Kristian Parey[1,2,3,7], Amina Djurabekova[4,8], Outi Haapanen [4,8], Karin Siegmund[1,2], Klaus Zwicker [5], Vivek Sharma [4,6✉], Volker Zickermann [1,2] & Heike Angerer [1,2✉]

Respiratory complex I catalyzes electron transfer from NADH to ubiquinone (Q) coupled to vectorial proton translocation across the inner mitochondrial membrane. Despite recent progress in structure determination of this very large membrane protein complex, the coupling mechanism is a matter of ongoing debate and the function of accessory subunits surrounding the canonical core subunits is essentially unknown. Concerted rearrangements within a cluster of conserved loops of central subunits NDUFS2 ($\beta$1-$\beta$2$^{S2}$ loop), ND1 (TMH5-6$^{ND1}$ loop) and ND3 (TMH1-2$^{ND3}$ loop) were suggested to be critical for its proton pumping mechanism. Here, we show that stabilization of the TMH1-2$^{ND3}$ loop by accessory subunit LYRM6 (NDUFA6) is pivotal for energy conversion by mitochondrial complex I. We determined the high-resolution structure of inactive mutant F89A$^{LYRM6}$ of eukaryotic complex I from the yeast *Yarrowia lipolytica* and found long-range structural changes affecting the entire loop cluster. In atomistic molecular dynamics simulations of the mutant, we observed conformational transitions in the loop cluster that disrupted a putative pathway for delivery of substrate protons required in Q redox chemistry. Our results elucidate in detail the essential role of accessory subunit LYRM6 for the function of eukaryotic complex I and offer clues on its redox-linked proton pumping mechanism.

[1] Medical School, Institute of Biochemistry II, Structural Bioenergetics Group, Goethe University, Max von Laue Str. 9, 60438 Frankfurt am Main, Germany. [2] Centre for Biomolecular Magnetic Resonance, Institute for Biophysical Chemistry, Goethe University, Max von Laue Str. 9, 60438 Frankfurt am Main, Germany. [3] Department of Structural Biology, Max Planck Institute of Biophysics, Max von Laue Str. 3, 60438 Frankfurt am Main, Germany. [4] Department of Physics, P.O. Box 64, University of Helsinki, FI-00014 Helsinki, Finland. [5] Medical School, Institute of Biochemistry I, Goethe University, Theodor-Stern-Kai 7, 60590 Frankfurt am Main, Germany. [6] HiLIFE Institute of Biotechnology, P.O. Box 56, University of Helsinki, FI-00014 Helsinki, Finland. [7] Present address: Department of Structural Biology, University of Osnabrück, Barbarastraße 13, 49076 Osnabrück, Germany. [8] These authors contributed equally: Amina Djurabekova, Outi Haapanen. ✉email: vivek.sharma@helsinki.fi; Angerer@em.uni-frankfurt.de

With a mass of ~1 MDa and more than 40 subunits mitochondrial NADH:ubiquinone oxidoreductase (complex I) is the largest enzyme complex of the respiratory chain[1–3]. Complex I is a redox-driven proton pump that contributes ~40% of the proton motive force driving ATP synthase. Dysfunction of respiratory complex I causes neuromuscular and neurodegenerative diseases[4]. Formation of reactive oxygen species (ROS) by complex I contributes to tissue damage in myocardial infarction[5,6]. The reversible active (A) to deactive (D) transition is thought to protect against reperfusion injury. Fourteen so-called central subunits of complex I are highly conserved from bacteria to human. Supposedly, they harbor all bioenergetic core functions of the enzyme complex. Electrons from NADH are accepted at the tip of the hydrophilic matrix arm and are transferred via a chain of Fe–S clusters to the ubiquinone (Q) reduction site close to Fe–S cluster N2 (Fig. 1). This site is connected with the membrane bilayer by a ~35 Å long tunnel for access of the hydrophobic substrate (Fig. 1 inset). Q reduction is coupled to proton pumping in the membrane arm of complex I, at a maximal distance of ~200 Å, which is remarkable at molecular length scales. We have proposed that a concerted rearrangement of three protein loops at the interface of membrane and matrix arm plays a key role in the coupling mechanism[7] and also possibly in the trapping and release of Q[8] (Fig. 1 inset). However, despite recent progress in structure determination and functional characterization of complex I, our understanding of redox-linked proton translocation at the molecular level is still fragmentary.

The function of accessory subunits surrounding the core of central subunits is largely uncharacterized. We have reported that deletion of the gene for accessory subunit NDUFA6 caused complete loss of Q reductase activity despite assembly of all central subunits[9], but molecular aspects of this inactivity have remained unclear. NDUFA6 belongs to the superfamily of LYRM proteins that typically form a heterodimer with the mitochondrial acyl carrier protein (ACPM, NDUFAB1) and bind the acyl cofactor, respectively[10–14]. Here we use the designation LYRM6 for this subunit. The LYRM6-ACPM heterodimer binds close to the functionally critical interface region of membrane and matrix arm of complex I (Fig. 1) and two adjacent loops of accessory LYRM6 interact with central subunits of mitochondrial complex I[1]. Here, we show by combining mutagenesis, structural biology, and molecular dynamics (MD) simulations the presence of a proton transfer pathway in this critical region connecting the bulk N phase with the active site. Our data suggest an essential function of accessory subunit LYRM6 in the controlled access of chemical protons for Q reduction in eukaryotic complex I.

## Results and discussion

**Site-directed mutagenesis of two loops of accessory subunit LYRM6.** In this work, we exchanged single residues in LYRM6 that contact central subunits ND3, NDUFS2, and NDUFS7 by site-directed mutagenesis (Supplementary Figs. 1, 2, Supplementary Table 1). For several mutants we observed decreased complex I activity while assembly of the enzyme complex was unperturbed (Supplementary Table 1, Supplementary Fig. 3). A substantial decrease of Q reductase activity was caused by exchange of LYRM6 residues interacting with residues in central subunits NDUFS2 (W90A$^{LYRM6}$) and NDUFS7 (Q92A$^{LYRM6}$) and accessory subunit NDUFA9 (E44A$^{LYRM6}$). The strongest impact on activity with residual rates below 25% was observed for mutants L42A$^{LYRM6}$, Y43A$^{LYRM6}$, and F89A$^{LYRM6}$ (Fig. 1 inset). Interestingly, these residues reside in the center of the accessory LYRM6 loop arrangement that interacts with a short section (L42-T43-S44$^{ND3}$) of the TMH1-2$^{ND3}$ loop that is downstream to the functionally highly critical and conserved E39-C40-G41$^{ND3}$ segment (see below) (Fig. 1 inset, Supplementary Fig. 2). Residue F89$^{LYRM6}$ of the accessory LYRM6 loop forms a prominent supporting structural element for the TMH1-2$^{ND3}$ loop (Supplementary Fig. 2). Analysis of purified mutant complex I by EPR spectroscopy excluded global assembly defects and loss or derangement of any EPR detectable Fe-S clusters (Supplementary Fig. 4). Overall, these results demonstrate that in three independent cases fully assembled complex I mutated in one position of accessory subunit LYRM6 showed dramatically decreased complex I activity.

**Cryo-EM structure of mutant F89A$^{LYRM6}$.** To understand the structural basis of this remarkable impact on eukaryotic complex I function, we determined the structure of inactive F89A$^{LYRM6}$ mutant by cryo-EM (Fig. 2, Supplementary Table 2).

Comparison of the 3.0 Å resolution map with our recent 3.2 Å resolution structure of the WT[15] showed that the matrix arm was tilted and density corresponding to specific structural elements in the interface region was missing indicating disorder (Fig. 2, Supplementary Figs. 5 and 6). This included residues A37-Q47$^{ND3}$ of the TMH1-2$^{ND3}$ loop, residues E208-E218$^{ND1}$ of the TMH5-6$^{ND1}$ loop, residues R219-K228$^{A9}$, R281-L289$^{A9}$ and D342-N359$^{A9}$ of accessory NDUFA9, and several lipids at the membrane interface (Fig. 2, Supplementary Fig. 5). Density for the β1-β2$^{S2}$ loop was visible but less clear precluding modeling of side chain positions for H91$^{S2}$, P92$^{S2}$ and H95$^{S2}$ (Fig. 2, Supplementary Fig. 5d). It is remarkable that a single mutation affecting a contact point of accessory subunit LYRM6 with the TMH1-2$^{ND3}$ loop causes structural changes in three central subunits and over a maximal distance of more than 50 Å. This indicates that the interface region between LYRM6 and NDUFA9

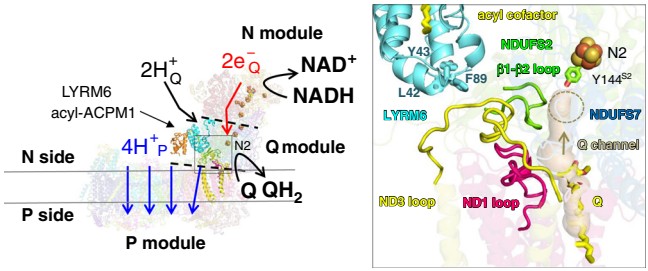

**Fig. 1 Conserved loop cluster of mitochondrial complex I formed by central subunits ND1, ND3, and NDUFS2.** Architecture of mitochondrial complex I from *Y. lipolytica* (PDB 6RFR). Cofactors of the central subunits (FMN and Fe-S clusters) and a bound Q molecule (Q) in the access pathway are highlighted (electron transfer pathway to Q is indicated by red arrow). The peripheral arm is composed of the N module (NADH oxidation) and the Q module (Q reduction). The membrane arm represents the proton pumping (P) module with the pathways (blue arrows) for pumped protons H$^+$$_P$. The reduction of Q requires electrons from terminal Fe-S cluster N2 and two protons for Q redox chemistry (H$^+$$_Q$, black arrow) from the negative side of the membrane. Interfaces between modules are indicated by black dashed lines. The accessory LYRM6/ACPM1 (NDUFA6/NDUFAB1-α) heterodimer at the Q/P module interface is highlighted. Subunits ND1, ND3, and NDUFS2 form an important part of the Q/P module interface region. Inset, the TMH1-2$^{ND3}$ loop of subunit ND3 (yellow) is located at the Q/P module interface and interacts with the N-terminal β-sheet$^{S2}$ of subunit NDUFS2 (green). The TMH5-6$^{ND1}$ loop (hot pink), THM1-2$^{ND3}$ loop (yellow) and β1-β2$^{S2}$ loop (green) form a conserved loop cluster. The tunnel for access of Q to the Q reduction site is shown in transparent brown color and the Q reduction site close to Y144$^{S2}$ is highlighted in brown dashed lines. Residues L42$^{LYRM6}$, Y43$^{LYRM6}$, F89$^{LYRM6}$, Y144$^{S2}$, the Q molecule and acyl cofactor are shown in stick representation, respectively.

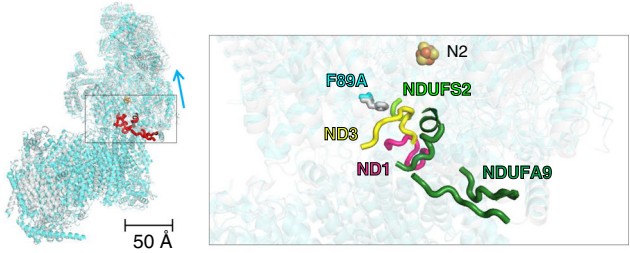

**Fig. 2 Weak or missing density in cryo-EM map indicates disorder at the interface of membrane and matrix arm of inactive mutant F89A[LYRM6].** Superimposed models of F89A[LYRM6] mutant (cyan) and WT complex I from *Y. lipolytica* (PDB 6RFR, gray) (superimposed at the central section of the membrane arm (subunit ND4); Fe-S cluster N2 is shown in sphere representation). Structural elements showing disorder in mutant F89A[LYRM6] are highlighted in red. Inset, residues at position 89[LYRM6] of subunit LYRM6 are shown in stick representation for WT and mutant. Structural elements of subunits ND1 (hot pink), ND3 (yellow), NDUFS2 (green) and accessory NDUFA9 (forest) showing disorder in mutant F89A[LYRM6] are displayed in cartoon representation (density maps are shown in Supplementary Fig. 5). The arrow indicates tilting of the peripheral arm in the mutant (Supplementary Fig. 6).

reacts as an interconnected structural unit and supports the proposed role of the TMH1-2[ND3] loop to transmit and drive structural changes in the interface loop cluster and active site. We have previously shown that the unfolded Q site model for the A/D transition developed for mammalian complex I[16] does not apply to the yeast enzyme[15]. Nevertheless, the changes observed in the F89A[LYRM6] mutant are comparable to the structural differences that were previously described between the A and D form of complex I from mouse[17] (Supplementary Fig. 7). This surprising similarity highlights the inherent flexibility of the interface region. However, in contrast to controlled and reversible relaxation of mouse complex I into the D form, disruption of a stabilizing interaction between LYRM6 and the TMH1-2[ND3] loop causes an irreversible decrease of activity in the *Y. lipolytica* LYRM6 mutant(s) (Supplementary Table 1).

Several lines of evidence have highlighted the critical role of the TMH1-2[ND3] loop. Upstream of the interaction site with LYRM6, the TMH1-2[ND3] loop harbors a strictly conserved EXG motif (E39[ND3]-G41[ND3]). Mutation of invariant E39[ND3] to alanine in bacterial complex I is known to drastically affect the Q reductase activity[18]. In addition, positions of human pathogenic mutations S34P[ND3], S45P[ND3], and A47T[ND3] (human complex I numbering) are close to the LYRM6/ND3 interface[19] (Supplementary Fig. 8). In mitochondrial complex I and many bacterial species the central residue of the EXG motif is a cysteine (C40[ND3]). Immobilization of the E39-C40-G41[ND3] segment of the TMH1-2[ND3] loop by an engineered disulfide bond in mutant Q133C[S7] (C40[ND3]-C133[S7]) was recently shown to disengage the proton pumps[20]. The residue corresponding to E39[ND3] in the eubacterium *Thermus thermophilus* was also suggested to reside at the entrance of a channel-like path for delivery of protons for Q redox chemistry close to Fe-S cluster N2[21] (Supplementary Figs. 2e and 9b). Note that conversion of Q to QH2 requires two electrons and two substrate protons and that the timing of coupled electron and proton transfer events is thought to be of fundamental importance for the coupling mechanism of complex I[22–24]. Uncontrolled protonation of reduced Q intermediates by bulk protons would result in heat production instead of controlled energy conversion for the generation of proton pumping strokes. Therefore, concerted (re-)protonation of anionic Q intermediates or highly conserved and functionally essential residues H91[S2], H95[S2] and Y144[S2] of the β1-β2[S2]

loop[25–28] from the bulk N phase via highly conserved E39[ND3] could be central to the proton pumping mechanism. Noteworthy, the TMH1-2[ND3] loop segments harboring the EXG motif were modeled in alternative conformations in *T. thermophilus* and *Y. lipolytica* complex I structures suggesting flexibility at this position[15,29] (Supplementary Fig. 9).

**CAVER tunnel calculations and mutagenesis of tunnel residues.** Our CAVER-based[30,31] structural analysis of WT *Y. lipolytica* complex I (PDB 6RFR) revealed a tunnel mainly formed by the flexible loops of central subunits ND1, ND3, NDUFS2 and by a helical element of NDUFS7, connecting the bulk N phase with the middle of the Q channel (Fig. 3a, Supplementary Table 3, Supplementary Fig. 10). Interestingly, the entrance of the tunnel was shaped by several residues of accessory subunits LYRM6 and NDUFA9, respectively (Supplementary Table 3). The tunnel had a length of 60 Å and a bottleneck radius of 1.27 Å. Highly conserved residue E39[ND3] of the TMH1-2[ND3] loop was part of the tunnel and the tunnel entrance was close to the LYRM6/ND3 interface (Fig. 3a). To evaluate the functional significance of the identified CAVER tunnel we exchanged ten residues of central subunits NDUFS2 and NDUFS7 to alanines by site-directed mutagenesis (Supplementary Table 3, Supplementary Fig. 10). Eight of ten mutants of the tunnel showed decreased complex I activity below 20% indicating functional relevance of the tunnel. CAVER analysis using our model of the F89A[LYRM6] mutant revealed disruption of the CAVER tunnel at position of the β1-β2[S2] loop (Supplementary Fig. 10).

**Multiscale modeling and simulations of WT and mutant enzymes.** In order to understand the dynamical aspects of putative proton access pathways and why point mutations in an accessory subunit affect Q reductase activity so drastically, we performed fully atomistic MD simulations of WT *Y. lipolytica* complex I, and F89A[LYRM6] and Y43A[LYRM6] mutants (Supplementary Table 4). In our simulations of WT enzyme, we identified a hydrated pathway at the interface of ND1, ND3 and NDUFS2 subunits, in similar location to the passage observed in our CAVER-based structural analysis (Fig. 3a, b). The hydrated channel in mitochondrial complex I involved several functionally important residues, also from the central loop cluster and active site of Q reduction (Fig. 3b, Supplementary Table 3), and might be used for substrate proton transfer. Notably, some of the residues identified by CAVER analysis and also by hydration analysis in simulations indeed show low Q reductase activity upon point mutations (Supplementary Table 3). Interestingly, in WT simulations functionally critical E39[ND3] stabilized ion pairing with the conserved K130[ND1] of the ND1 subunit (Fig. 3f), an interaction observed in two structures of mammalian complex I[17,32] (Supplementary Fig. 9). When we modeled a hydronium ion ($H_3O^+$) in the hydrated channel, it rapidly transferred to the anionic E39[ND3] via Grotthuss-like proton transfer on water wires in multiple independent QM/MM (quantum mechanical/molecular mechanical) MD simulations of WT enzyme (Supplementary Table 6, Supplementary Fig. 11). Protonation of E39[ND3] did not occur in F89A[LYRM6] mutant simulation, whereas when E39[ND3] was modeled neutral in WT setup, proton remained stable on it. Furthermore, rapid protonation of functionally critical H91[S2] from E39[ND3] was found to occur in a QM/MM MD run when the two residues are hydrogen bonding (Supplementary Fig. 11). These simulations of explicit proton transfer support the view that the channel could be used to supply substrate protons via the E39[ND3]-H91[S2] pathway to the Q reduction site in a gated manner (see below).

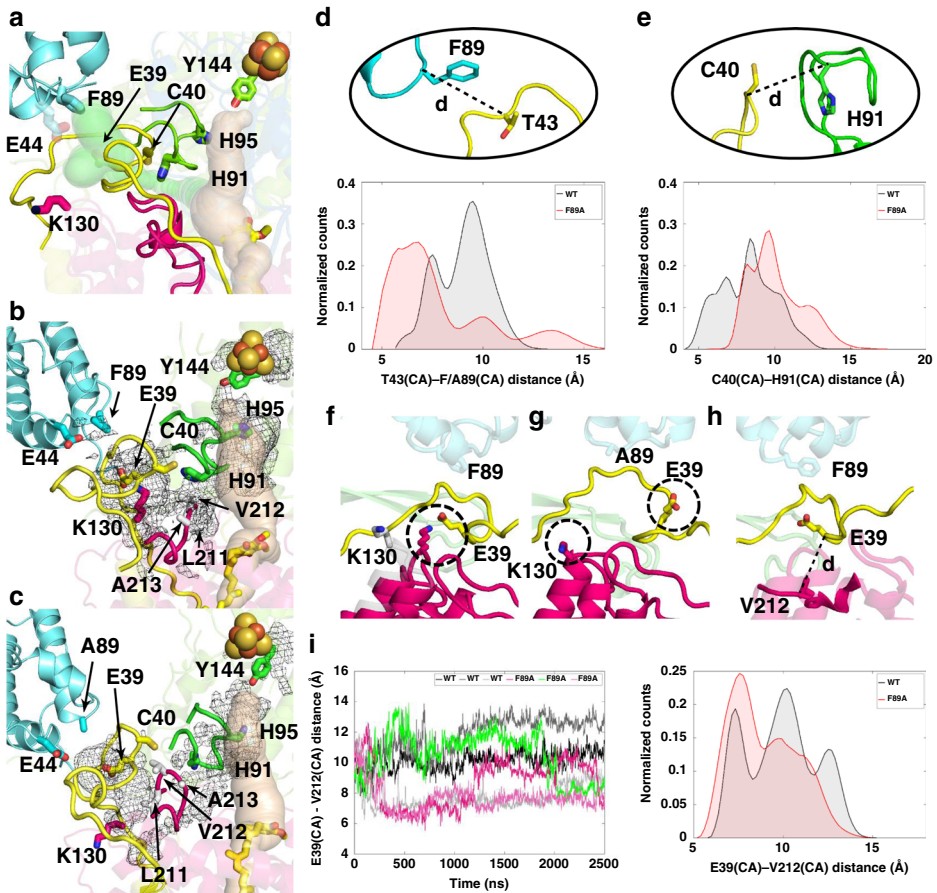

**Fig. 3 Dynamics of key segments at the interface of ND3 and LYRM6 subunits.** (**a**) shows the Q binding cavity (beige) and the tunnel identified (light green) in WT *Y. lipolytica* complex I structure (PDB 6RFR) using CAVER software[31] (radius 1.27 Å). Residue E44$^{LYRM6}$ highlights the entrance of the tunnel. Hydration of the proposed proton transfer path is shown by a dark gray mesh (**b**) calculated by averaging water occupancy in WT simulations. In contrast, in F89A$^{LYRM6}$ mutant MD simulations, the hydrated path is blocked by conserved hydrophobic residues (L211$^{ND1}$ and V212$^{ND1}$) from the ND1 subunit (**c**) due to structural rearrangements. The absence of the bulky side chain in the F89A$^{LYRM6}$ mutant causes instability in the ND3 segment and its displacement towards the LYRM6 segment (**d**), and away from the β1-β2$^{S2}$ loop (**e**). The selected distances are shown as histograms calculated from simulation trajectories. See also Supplementary Table 5 for similar data on complex I structures. The salt-bridge (K130$^{ND1}$-E39$^{ND3}$) is stabilized in WT simulation (**f**) in contrast to F89A$^{LYRM6}$ mutant simulations (**g**); gray, conformation of K130$^{ND1}$ in WT structure (PDB 6RFR). Shift of the short hydrophobic stretch from ND1 loop segment (L211-V212-A213-G214$^{ND1}$) blocks the channel, as shown by time series and histogram of distance between E39$^{ND3}$ and V212$^{ND1}$ (**h, i**). Subunit and amino acid coloring as in Fig. 1 along with selected residues shown in sticks and labeled. The isovalue of the mesh is 0.2 (20% water occupancy) in panels (**b**) and (**c**), and two simulation replicas were used to calculate hydration (r1 and r2 in WT and r1 and r3 in mutant, see also Supplementary Figs. 15, 16). In panels (**d**, **e,** and **i**), the histograms show the distribution of distances normalized so that the area under curve equals 1.

In classical MD simulations of the F89A$^{LYRM6}$ mutant, the conformation of the backbone of the L42-T43-S44$^{ND3}$ segment that brushes the ND3 loop was clearly different from that of the WT enzyme. Due to the space created by the smaller side chain at position 89$^{LYRM6}$ in LYRM6, the TMH1-2$^{ND3}$ loop segment was found to be much more flexible and relaxed towards LYRM6, and away from the β1-β2$^{S2}$ loop (Fig. 3d, e). A similar, albeit weaker effect, was also seen in the simulations of the Y43A$^{LYRM6}$ mutant (Supplementary Fig. 12). When F89A$^{LYRM6}$ simulations were performed with H91$^{S2}$ protonated, the shift in ND3 loop segment was observed, whereas in WT case it remained relatively stable (Supplementary Fig. 12). The mobility and structural displacement of the TMH1-2$^{ND3}$ loop segment observed in several independent mutant simulations is in agreement with the cryo-EM data on the F89A$^{LYRM6}$ mutant that showed the absence of density for the ND3 segment involving residues A37-Q47$^{ND3}$ (Supplementary Fig. 5d). To compare the dynamics of the TMH1-2$^{ND3}$ loop with bacterial enzymes that lack accessory subunits, we performed atomistic MD simulations on the complex I structure from *T. thermophilus* (PDB 4HEA) and also on the large-scale model of

complex I from *Y. lipolytica* (PDB 6RFR, Supplementary Table 4). The data revealed a higher level of mobility in the TMH1-2$^{ND3}$ loop in bacterial complex I compared to the eukaryotic enzyme (Supplementary Fig. 13). The observed difference in loop dynamics most likely originates from the absence of LYRM6 accessory subunit in the bacterial enzyme even though the two enzymes (*Y. lipolytica* and *T. thermophilus* complexes) differ in several aspects such as subunit composition and sequence variation. This suggests that the LYRM6 subunit not only provides scaffolding to the functionally important TMH1-2$^{ND3}$ loop, but also restricts its dynamics to allow functionally coordinated movements (Fig. 3). We propose that bacterial enzymes utilize the same proton transfer pathway, but due to the lack of coordinated movement of ND3 loop, the protonation of anionic intermediates at the Q redox site is controlled differently.

Interestingly, the conformation of the ND3 loop segment observed in F89A$^{LYRM6}$ mutant simulations prevented stabilization of the K130$^{ND1}$-E39$^{ND3}$ ion-pair (Fig. 3g) and also led to the creation of a barrier by a short stretch of hydrophobic residues L211-V212-A213-G214$^{ND1}$ from the TMH5-6$^{ND1}$ loop blocking

the hydrated channel observed in WT simulations (Fig. 3c). We note that a similar obstruction of the channel is seen in bovine complex I[33] (Supplementary Fig. 14) formed by residues L207-V208[ND1] analogous to L211-V212[ND1] in *Y. lipolytica* complex I (Supplementary Table 5). Exchange of the corresponding valine residue in human complex I is a well-known pathogenic mitochondrial mutation[34] (Supplementary Fig. 8). Missing density for the L211-V212[ND1] segment in the F89A[LYRM6] mutant map precluded its structural modeling; however, simulations supported the existence of differently populated conformational substates (Fig. 3). Conformational flexibility, as seen in F89A[LYRM6] cryo-EM experiment, can also be interpreted as additional conformational states visited by TMH5-6[ND1] and TMH1-2[ND3] loops, which are less populated in WT enzyme. In our simulations, a correlation between V212[ND1]-E39[ND3] distance and channel hydration was found (Supplementary Figs. 15 and 16). A shorter V212[ND1]-E39[ND3] distance, as observed in F89A[LYRM6] mutant simulations, blocked the channel hydration in the region, whereas larger V212[ND1]-E39[ND3] distance was associated with a higher level of hydration (Fig. 3 and Supplementary Figs. 15 and 16). We suggest that the F89A[LYRM6] induced change in conformation of the conserved loop cluster (TMH1-2[ND3], β1-β2[S2] and TMH5-6[ND1] loops) closes the hydrated path observed in WT simulations, and thus prevents the transfer of substrate protons required for Q reduction chemistry from the N phase of the membrane (Supplementary Fig. 17). The conserved hydrophobic segment from functionally critical TMH5-6[ND1] loop (Supplementary Table 5) may thus function as a necessary gating element for controlled transfer of substrate protons. In contrast to the F89A[LYRM6] mutant, simulations of the uncoupled Q133C[S7] mutant (with C40[ND3]-C133[S7] disulfide bond modeled) showed no blockage by the TMH5-6[ND1] segment and formation of a hydrated path similar to WT (Supplementary Fig. 12). The arrest of the TMH1-2[ND3] loop by the C40[ND3]-C133[S7] disulfide bond in the previously described Q133C[S7] mutant[20] would preclude closing of the proposed substrate proton pathway. Thus, an uncontrolled flow of protons from the bulk to the Q reduction site might explain uncoupling of redox chemistry from proton pumping in the Q133C[S7] mutant by decreasing the lifetime of negatively charged reaction intermediates[28].

Taken together, we demonstrate that a single point mutation in the accessory LYRM6 subunit has a significant impact on the energy conversion reactions of mitochondrial complex I. Our high-resolution structure of mutant F89A[LYRM6] and MD simulations show that structural perturbations originating at a contact point of LYRM6 with the TMH1-2[ND3] loop propagate to functionally critical loops of central subunits at the interface of membrane and matrix arm. This indicates that amino acid residues from the accessory subunit provide indispensable stabilizing interactions for the efficient functioning of mitochondrial complex I. Our combined structural and simulation data suggests the presence of an entry pathway for protons consumed in Q redox chemistry, and that the control of proton access to the active site is critical for the coupling mechanism of complex I.

While this paper was under revision, two high-resolution cryo-EM structures of mitochondrial complex I were published[35,36]. Based on conserved positions of water molecules, Grba and Hirst[35] suggested a pathway for substrate proton transfer in the peripheral arm close to the Q reduction site of complex I. Kampjut and Sazanov[36] on the other hand suggested the uptake of chemical protons from the bulk N phase into core hydrophobic subunit ND4L of the membrane arm. The two proposed pathways for chemical protons are strikingly different from our findings as our combined biochemical, structural, and computational data indicate that the central loop cluster is the structural element that provides proton access to the active site in a controlled fashion. Further work is needed to unravel the functional significance of each proposed proton transfer pathway.

## Methods

**Site-directed mutagenesis of complex I subunits from *Y. lipolytica*.** Point mutations were generated by inverse PCR mutagenesis using the plasmids pUB26-WT-*nb4m(lyrm6)*, pUB26-WT-*nucm* and pUB4-WT-*nukm*. After sequencing, the plasmids were transformed into *Y. lipolytica* strain Δ*nb4m/lyrm6*, Δ*nucm* or Δ*nukm*[9,26]. Plasmid DNA was transformed into *Y. lipolytica* cells using the lithium acetate method. Cells from 0.5–1 ml of an overnight culture were spun down, washed with 0.9% NaCl and centrifuged again. The cells were re-suspended in 90 μl of 50% PEG4000 in the presence of 100 mM lithium acetate pH 6, 100 mM DTT and 0.25 mg/ml ssDNA. 200–300 ng of plasmid DNA were added for each transformation mixture, vortexed and incubated at 39 °C for 60 min. After transformation, the cells were plated on the appropriate selection medium and incubated for 3–4 days at 28 °C.

**Preparation of mitochondrial membranes and purification of complex I.** *Y. lipolytica* WT (GB30) strain and the strains complemented with the plasmid carrying the WT genes were grown in YPD medium with or without hygromycin, respectively. Mitochondrial membranes were prepared as described previously[26]. For small scale preparation of mitochondrial membranes, pellets of 5–8 g of cells (wet weight) in 50 ml falcon tubes were re-suspended in 10 ml mito-buffer (600 mM sucrose, 20 mM Mops, 1 mM EDTA, pH 7.2) supplemented with 2 mM PMSF. Cells were broken by vortexing in a series of 15 × 1 min with 1 min resting intervals on ice in the presence of 10 g glass beads (0.25–0.5 mm). The resulting suspension was filtered through gauze and filled up to 30 ml with the mito-buffer. Cells debris and glass beads were removed by centrifugation at 3238 × g (4000 rpm, Megafuge, rotor 2704) for 30 min at 4 °C. Supernatants were collected and mitochondrial membranes were sedimented by ultracentrifugation at 147,642 × g (35,000 rpm, rotor Ti 50.2) for 1 h at 4 °C. Finally, the supernatant was discarded, and pellets were re-suspended in 0.5–1 ml mito-buffer. The suspension was then homogenized on ice in the presence of 5 mM PMSF using a glass homogenizer. Aliquots of the membrane suspension were immediately frozen in liquid nitrogen and stored at −80 °C. For purification of complex I, large-scale preparations of mitochondrial membranes were performed. Up to 200 g of cells (wet weight) of *Y. lipolytica* harvested after fermentation were re-suspended in 400 ml mito-buffer. Cells were broken using a cooled Cell-Desintegrator-C (Bernd Euler Biotechnologie) and 80 ml of 0.5 mm glass beads for 2 h. Cells debris was removed by centrifugation with 4416 × g (5000 rpm; rotor JA10) for 30 min at 4 °C. The supernatant was filtered through gauze and diluted with the mito-buffer up to a final volume of 450 ml. Mitochondrial membranes were obtained by ultra-centrifugation with 167,425 × g (38,000 rpm, rotor Ti 45) for 1.5 h at 4 °C. Pelleted membranes were re-suspended in mito-buffer without EDTA and centrifuged 174,040 × g (38,000 rpm, rotor Ti 50.2) for 1 h at 4 °C. Pelleted membranes were homogenized and the membrane suspension was supplemented with 20 mM Na-borate pH 7.9, 50 mM NaCl and 1.5 mM PMSF. Aliquots were frozen in liquid nitrogen immediately after preparation and stored at −80 °C.

Complex I was solubilized with n-dodecyl-ß-D-maltoside (DDM) as detailed in[37]; however, purification buffers contained 0.025% 2,2-didecylpropane-1,3-bis-β-D-maltopyranoside (LMNG) instead of DDM. Mitochondrial membranes were thawed and diluted to a concentration of 18 mg/ml protein with 50 mM NaCl, 20 mM Na-borate; pH 7.9, 1.5 mM PMSF. DDM was added dropwise with stirring on ice to a final detergent to protein ratio of 1g:1g. The solution was further stirred on ice for 10 min and then centrifuged at 147,642 × g (35,000 rpm; rotor Ti50.2) for 1 h at 4 °C. The supernatant was adjusted to 400 mM NaCl, 55 mM imidazole, 0.8 mM MgCl₂ and pH 7.3–7.4 and loaded onto a Ni-NTA sepharose column (Bio-Rad), which was prior equilibrated with buffer A (55 mM imidazole, 400 mM NaCl, 0.025% LMNG, and 20 mM Na-phosphate, pH 7.2). The presence of the 6xHis tag attached to the 30 kDa subunit of the peripheral arm of complex I allows the enzyme to bind. After washing the column with 150 ml of the same buffer (buffer A), complex I was eluted with buffer B (200 ml of 140 mM imidazole, 400 mM NaCl, 0.025% LMNG, and 20 mM Na-phosphate, pH 7.2) containing a higher imidazole concentration. Fractions with a HAR activity (see below) higher than 20 U/ml were combined and concentrated using Centriped centrifugal filter devices (Millipore®). To remove imidazole and to optimize purity of the complex I sample, the concentrated pool was applied to a TSK G4000SW column (TosoH Bioscience) and separated by size exclusion chromatography on an ÄKTA purifier chromatography system (GE Healthcare). Equilibration and elution were performed with FPLC buffer (100 mM NaCl, 1 mM EDTA, 20 mM Tris/HCl, pH 7.2 and 0.025% LMNG). The process was monitored by UV absorption at 280 nm characteristic for protein and 415 nm characteristic for the FMN, which is a cofactor of complex I. Peak fractions were pooled and concentrated using spin devices (Vivaspin, 100,000 MWCO, Sartorius). Aliquots were shock frozen and stored in liquid nitrogen.

Specific NADH:hexaammineruthenium(III) (HAR) and 2-decyl-4-quinazolinylamine (DQA) inhibitor-sensitive deamino-NADH:decyl-ubiquinone (dNADH:DBQ) oxidoreductase activity were measured for mitochondrial membranes. NADH:HAR and NADH:DBQ activities of purified enzymes were measured in the presence of 0.01% LMNG. Complex I content in mitochondrial membranes as well as in purified enzyme is typically determined as NADH:HAR

oxidoreductase activity. This unphysiological reaction depends on the presence of an intact NADH oxidation site in the N-module of complex I and reflects the amount of fully assembled complex I in a preparation. The reaction was monitored spectrophotometrically at 340 nm wavelength ($\varepsilon = 6.22$ mM$^{-1}$ cm$^{-1}$) for a decrease in absorbance proportional to NADH oxidation. The reaction was started by the addition of 20–40 µg protein of mitochondrial membranes or 1–2.5 µg of purified complex I to the HAR buffer (250 mM sucrose, 20 mM HEPES, 0.2 mM EDTA, 2 mM KCN, ±0.01% LMNG; pH 8) containing 200 µM NADH and 2 mM HAR. Activity measurements were performed at 30 °C in a Shimadzu UV-2450 spectrophotometer and all values were normalized to the protein concentration. For each mutant two biological replicates were analyzed, and each measurement was carried out in duplicate. Complex I activities in mitochondrial membranes as well as in purified enzyme were determined by measuring the physiological electron transfer activity of NADH to ubiquinone. Short chain ubiquinone analogues were used to avoid solubility problems due to the isoprene chain of Q9, which is the natural electron acceptor of complex I from *Y. lipolytica*. To exclude NADH oxidation by alternative dehydrogenase (NDH2) in mitochondria of *Y. lipolytica*, deaminoNADH (dNADH) instead of NADH was used during the activity assays. dNADH is a NADH analogue in which the adenine moiety is replaced by hypoxanthine. The reaction was started by the addition of 100 µM dNADH/NADH to a solution of DBQ buffer (20 mM MOPS-Na; 50 mM NaCl; 2 mM KCN, ±0.01% LMNG; pH 7.2) containing 60/150 µM DBQ and 20–40 µg protein of mitochondrial membranes or 2–5 µg of purified complex I. The reaction was stopped by the addition of 2 µM complex I inhibitor 2-decyl-4-quinazolinylamine (DQA). The inhibitor-insensitive activity was subtracted from the initial measurement and the result was normalized to complex I content to allow comparison between different preparations. All activities measurements were performed at 30 °C in a Shimadzu UV-2450 spectrophotometer. For each mutant two biological replicates of mitochondrial membranes were analyzed, and each measurement was carried out in duplicate. Source data are provided as a Source Data file.

**Electrophoresis**. Mitochondrial membranes from *Y. lipolytica* were solubilized by DDM and separated by blue native electrophoresis (BN-PAGE) with an 4–16% acrylamide gradient[38]. Briefly, mitochondrial membranes (400 µg protein) were first diluted in 300 µl of water and sedimented by centrifugation for 10 min at 16,100 × *g*. The membrane pellets were re-suspended with 40 µl solubilization buffer A (50 mM imidazole/HCl; 50 mM NaCl; 2 mM 6-aminohexanoic acid; 1 mM EDTA; pH 7.0). 1.5 g DDM per g protein was added to the sample and left to solubilize for 10 min on ice. The solubilized membranes were centrifuged at 16,100 × *g* for 45 min at 4 °C and supernatants were supplemented with 5% Coomassie brilliant blue G-250 and 5 µl of 50% glycerol. 20 µl samples were then loaded on 4–16% acrylamide gradient gels (Bio-Rad) and protein complexes were separated for about 4 h at 16 mA and 400 V in the cold room (4 °C). Uncropped gel is provided as a Source Data file.

**EPR Spectroscopy**. X-band EPR spectra were obtained with a Bruker ESP 300E spectrometer equipped with an HP 53159A frequency counter (Hewlett Packard), ER 035M NMR gaussmeter (Bruker, BioSpin), and a liquid helium continuous flow cryostat (Oxford Instruments). Spectra were recorded using the following parameters: microwave frequency 9.47 GHz, microwave power 1 mW, modulation amplitude 0.64 mT, and modulation frequency 100 kHz. Samples were reduced with 2 mM NADH. For comparison spectra were normalized on signal intensities of $g_x$N3 and $g_x$N4. Spectra were recorded at 12 K to analyze binuclear and tetra-nuclear clusters. Samples were frozen in cold isopentane/methylcyclohexane (5:1, 120 K) and stored in liquid nitrogen.

**Structure modeling and CAVER analysis**. For structural modeling we employed the bioinformatic tool PyMOL[39] using the pdb entry files PDB 6RFR (WT *Y. lipolytica*[15], PDB 6G2J (mouse A form[17], PDB 6H8K (Q133C[S7] *Y. lipolytica*[20], PDB 5LNK (sheep complex I[14], PDB 6QC5 (sheep closed form[32], PDB 6QBX (sheep closed form[32], PDB 4HEA (*T. thermophilus*[29], PDB 5LC5 (bovine complex I[33], PDB 5XTH (human complex I[40], PDB 5XTD (human complex I[40], PDB 6HUM (photosynthetic complex I[41] and PDB 6CFW (membrane bound hydrogenase, MBH[42]. CAVER 3.0[31] was used as PyMOL[39] plugin. The critical Y144[S2] near cluster N2 served as initial starting point for tunnel calculation. The input model comprised the central subunits ND1, ND3, NDUFS2, NDUFS7 and the accessory subunits LYRM6 (NDUFA6), ACPM1 (NDUFAB1) and NDUFA9 from *Y. lipolytica* complex I. The minimum probe radius (bottleneck radius) varied from 1.2 Å to 1.5 Å and correspond to the narrowest part of a given tunnel. All other parameters maintained default.

**Cryo-EM structure**. Complex I F89A[LYRM6] mutant was purified in LMNG as detergent and polished by ion exchange chromatography using a MonoQ column. The sample was applied at a concentration of 1.5 mg/ml to freshly glow-discharged 1.2/1.3 holey carbon grids (Protochips). Grids were automatically blotted for 12–14 s in a Vitrobot Mark IV (Thermo Fisher Scientific Inc., USA) at 10 °C and 70% humidity (drain and wait time 0 s, blot force −2 a.u. (arbitrary units)), plunge-frozen in liquid ethane and stored in liquid nitrogen until further use. Cryo-EM data were collected automatically using EPU software (Thermo Fisher Scientific Inc.) on FEI

Titan Krios microscope (Thermo Fisher Scientific Inc.) at 300 kV equipped with K3 Summit detector (Gatan) operating in counting mode. Cryo-EM images were acquired at a nominal magnification of 105,000x with a calibrated pixel size of 0.837 Å, a defocus range from −1.5 to −2.5 µm, an exposure time of 3 s, and a total electron exposure on the specimen of ~51.8 e$^-$/Å$^2$. A set of 2016 dose-fractionated micrographs were subjected to motion correction and dose-weighting of frames by MotionCor2[43]. The micrograph-based contrast transfer function (CTF) was determined by Gctf and the resulting images were used for further analysis with the software package RELION3.0[44]. Particles were picked using Autopick within RELION3.0 and a total of 479,372 particles were extracted with a box size of 400 × 400 pixels. The extracted particles were subjected to reference-free 2D classification, which was performed with four-fold binned data to remove false positives and imperfect particles. A further 3D classification with a previous cryo-EM map of WT *Y. lipolytica* complex I[15] as an initial reference was applied. The best 3D class of 143,203 particles was used for auto-refinement, CTF refinement and Bayesian polishing in RELION3.0[44]. The final map at 2.96 Å resolution was sharpened using an isotropic B-factor of −65 Å$^2$ and the local resolution was estimated with ResMap (http://resmap.sourceforge.net)[45] (Supplementary Fig. 5). All resolutions were estimated using the 0.143 cutoff criterion[46] with gold-standard Fourier shell correlation (FSC) between two independently refined half maps[47]. The cryo-EM structure of WT *Y. lipolytica* complex I (PDB 6RFR) was used as template and critical areas were rebuilt with COOT[48]. The structure was refined using *phenix.real_space_refine* in combination with rigid-body refinement[49]. A quality check indicated excellent stereochemistry with 94.0% of the non-glycine and non-proline residues found in the most-favored regions and 0.06% outliers (all-atom clashscore: 8.07). Refinement and validation statistics are summarized in Supplementary Table 2.

**Atomistic molecular dynamics simulations**. We performed atomistic classical molecular dynamics (MD) simulations of complex I from *Y. lipolytica* (PDB 6RFR) (see also[50]). We did not use the F89A[LYRM6] mutant structure because of the missing density in the ND3/LYRM6 region, which would preclude unambiguous modeling of protein backbone and sidechains. The MD model systems included core subunits NDUFS2, NDUFS7, NDUFS3, NDUFS8, ND1, ND3, ND6, ND4L, ND2, and ND5 (terminal helix), as well as accessory subunits LYRM6 (NDUFA6), NDUFAB1, NDUFA12, NDUFA5, and NDUFA9. The selected subunits and cofactors fully encompass the region of interest (Fig. 3a), and smaller model systems were found stable (see below). In test simulations, we observed some of the N and C terminals of protein chains to be highly flexible (residue 1–54 NDUFS7, 36–52 and 256–272 in NDUFS3, 128–137 in NDUFA12, 40–65 in NDUFS8). Therefore, these were removed in order to maintain the small size of the simulation box. The structurally resolved quinone (Q9) molecule was included in the simulations. Standard protonation states of all amino acids were used, except H95[S2], which was kept protonated in all the simulations (see also Supplementary Table 4 for H91[S2] protonation states). The protein system was embedded into a pure POPC lipid membrane obtained from CHARMM-GUI[51–56] and aligned by OPM alignment[57]. The membrane-protein system was solvated with TIP3 water molecules and 0.1 M of Na$^+$ and Cl$^-$ ions. CHARMM force field was used for all components of the simulation setup; protein, lipids, solvent[58–60], iron-sulfur clusters[61], ZMP[53], and quinone[62]. The total system size was ca. 452,000 atoms (see Supplementary Fig. 18). The entire protein-membrane-solvent system was energy minimized followed by two equilibration steps in which constant temperature (310 K) and pressure (1 atm) were achieved with Berendsen thermostat[63] and Parrinello-Rahman barostat[64,65], respectively. In the first equilibration step (100 ps NVT and 1 ns NPT), 2000 kJ mol$^{-1}$ nm$^{-2}$ constraints were applied on the heavy atoms of the protein but keeping the ligands free. In the second step, all constraints were released, and an energy minimization procedure was followed by a 100 ps NVT and 1 ns NPT equilibration MD run. During production simulations, the temperature and pressure were kept at 310 K and 1 atm, respectively, using Nose-Hoover thermostat (time constant 1 ps)[66,67] and Parrinello-Rahman barostat (time constant 5 ps)[64,65]. The time step of the simulation was 2 fs, which was achieved by using the LINCS algorithm[68] and long-range electrostatics (12 Å cutoff) was dealt with Particle-mesh Ewald method[69] implemented in GROMACS (with van der Waals cutoff 12 Å).

To shed light on the stability of protein models simulated and to compare our results with bacterial enzyme that lacks accessory subunits, we also performed atomistic MD simulations on full structures of respiratory complexes from *Y. lipolytica* (PDB 6RFR) and *T. thermophilus* (PDB 4HEA) (see setups 7–9 in Supplementary Table 4). For modeling setup of large-scale models, see[15,70]. We find despite removing core/accessory subunits in WT and mutant models (setups 1–6), the systems remained stable as shown by root mean square displacement (Supplementary Fig. 18) and secondary structure analysis (Supplementary Fig. 19). All MD simulations were performed with simulation software GROMACS (versions 2018.6 and 2020.2)[71–73]. The simulation trajectories were analyzed with software VMD (1.9.3–1.9.5)[74], PyMOL (2.3)[39] and CAVER (3.0)[30,31,75].

**Quantum mechanical/Molecular mechanical (QM/MM) simulations**. We performed hybrid QM/MM MD simulations on selected classical MD snapshots to study the proton transfer reactions in the identified proton transfer pathway. We used QCHEM and CHARMM software[51,76], B3LYP density functional[77–80] with 6–31 G* basis set as implemented in QCHEM for QM/MM simulations, which were based on additive electrostatic embedding framework. The classical (MM) region comprised

hydrated ND6, ND3, NDUFA9, LYRM6, ND1, NDUFS7, and NDUFS2 subunits, whereas the QM region consisted of selected amino acid residues, water molecules, and hydronium ion. The model systems and simulation lengths are described in Supplementary Table 6. The boundary between QM and MM regions was treated by link atoms introduced between Cα and Cβ atoms of selected amino acid residues. Initially, classical energy minimization was performed on the whole system for 1000 steps. This was followed by a 200 step QM/MM energy minimization, prior to the start of QM/MM MD runs. The QM/MM MD simulations were performed at 310 K with 1 fs time step and dispersion corrections[81].

**Reporting summary.** Further information on research design is available in the Nature Research Reporting Summary linked to this article.

## Data availability

The cryo-EM structures of complex I mutant F89A^LYRM6 has been deposited in the PDB with PDB ID 6Y79 (Cryo-EM structure of a respiratory complex I F89A mutant) and the respective cryo-EM maps in the EMDB under accession numbers EMD-10711 (Cryo-EM structure of a respiratory complex I F89A mutant). All data needed to evaluate the conclusions in the paper are present in the paper and/or the Supplementary Information. Additional data related to this paper may be requested from the authors. Source data are provided with this paper.

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

## Acknowledgements
The authors thank I. Dikic for institutional support at Medical School, Institute of Biochemistry II, Frankfurt am Main and they thank W. Kühlbrandt for institutional support and Ö. Yildiz and J. F. Castillo-Hernández for computational support at the MPI of Biophysics. The authors wish to acknowledge CSC—IT Center for Science, Finland, for computational resources, including Pilot Grand Challenge projects complexIty (puhti supercomputer) and complexIty2 (mahti supercomputer). They also acknowledge PRACE for awarding us access to MareNostrum at Barcelona Supercomputing Center (BSC), Spain and to Marconi100 at CINECA, Italy. This work was supported by the Deutsche Forschungsgemeinschaft (grant AN 1080/3-1 to H.A. and ZI 552/4-2 to V.Z.). V.S. acknowledges research funding from the Academy of Finland (294652), the University of Helsinki, the Magnus Ehrnrooth Foundation and the Sigrid Jusélius Foundation. O.H. acknowledges research funds from the CHEMS doctoral school of the University of Helsinki. EM data were collected and processed at the cryo-EM facility of the MPI of Biophysics, funded by the Max Planck Society.

## Author contributions
E.G.Y. prepared and analyzed mitochondrial membranes, analyzed the data, performed structural analyses and drew the figures. K.P. prepared cryo-EM grids, acquired and processed cryo-EM data, built the model, analyzed the data and drew the figures. A.D. performed the QM/MM and classical MD simulations, analyzed the simulation data, and drew the figures. O.H. performed the classical MD simulations, analyzed the data and drew the figures. K.S. purified and characterized complex I mutants. K.Z. performed EPR spectroscopy, analyzed the data and drew the figure. V.S. analyzed the simulation data and interpreted its mechanistic implications, supervised the modeling and simulation work and wrote the manuscript. V.Z. interpreted the mechanistic implications of the structure and simulations, supervised the structural analyses, and wrote the manuscript. H.A. designed the study, prepared and analyzed mitochondrial membranes and purified complex I mutants, analyzed the data, performed structural analyses and drew figures, interpreted the mechanistic implications of the structure and simulations and wrote the manuscript.

## Funding

## Competing interests
The authors declare no competing interests.
