## [Peer Review File · Nature Communications]

REVIEWER COMMENTS

Reviewer #1 (Remarks to the Author):

The manuscript "Essential role of accessory subunit LYRM6 in the mechanism of mitochondrial complex I" by E.G. Yoga and colleagues discusses the role of accessory subunit LYRM6 in the complex I redox pathway. The authors have determined the electron cryo-microscopy structure of F89A LYRM6 mutant of complex I at 3.0 Å and also performed fully atomistic MD simulations of WT and F89A LYRM6 mutant complex I. Their analysis of the structure reveals how a single substitution F89A in LYRM6 results in long-range structural changes that affect the loop cluster formed by core subunits ND1, ND3 and NDUFS2. In conclusion the paper discusses how LYRM6 stabilizes the loop cluster that may act as a single structural unit to control the flow of protons required in Q redox chemistry through a newly discovered proton access pathway.

The manuscript is well written and clear.

Major comments:

The authors have previously shown that LYRM6 is essential for complex I activity (PMID: 24706851). The current manuscript "Essential role of accessory subunit LYRM6 in the mechanism of mitochondrial complex I" does not provide significant further insight. The observations made are mostly descriptive and conclusions drawn lack thorough experimental evidence. The observations lead the authors to make predictions about a specific mechanism linking LYRM6 to complex I turnover but these predictions are not tested. Also key observations that have been previously made from the authors' own work are omitted from the discussion, these observations significantly impact on the interpretation of the results presented here.

For instance, the authors have previously solved the structure of *Y. lipolytica* complex I during turnover in which they show disorder in the ND3 loop and many of the other loops and regions discussed here (eLife 2018;7:e39213, not referenced in this current manuscript). The flexible ND3 loop in the current structure therefore is not sufficient to explain reduced complex I activity. The two claims of "disorder means not functional" and "disorder during normal turnover" made in this and the previous work appear to be at odds and this contradiction needs to be addressed.

The authors argue that LYRM6 is necessary to hold the loop cluster formed by the core subunits in place to enable transport of protons required for Q redox chemistry. However, they do not discuss what holds the loop cluster in place in bacterial enzyme where a homologue of LYRM6 is absent? The authors should consider carrying out fully atomistic MD simulations of the bacterial enzyme, which lacks LYRM6, and comparing it to their current simulations.

Insufficient justification is given for the choice of residues mutated in this study. If they are interested in conserved features of complex I regulation by LYRM6, why did they not target conserved residues? It is unclear why this poorly conserved set of residues were targeted. If they were structure based what were the specific criteria used?

The authors claim that F89A, Y43A substitution reduces complex I activity via interaction with the ND3 loop residues. However, they do not show any details of the specific interactions with between LYRM6 and the ND3 loop. If they want to demonstrate the importance of these interactions, they should consider carrying out complementary mutations substituting the residues interacting with F89 and Y43 in the loop cluster and studying the effect of those substitutions on complex I activity. In other words, the authors need to make a prediction and test that prediction to prove their hypothesis, otherwise the study is merely descriptive and we already know LYRM6 is important for activity.

The authors should discuss how the Y43A mutant would result in decreased activity of the complex. Y43 is positioned to avoid any steric clashes with the loop.

The ND1 loop that "blocks" the hydration path in the simulation of the mutant is disordered in the structure indicating conformational flexibility. Flexibility calls into question the degree to which the mutant stabilizes this "blocking" conformation of the loop. How do the authors reconcile these contradictory results?

The authors don't show convincingly the mechanism by which this mutant is disrupting activity. According to their own work, disorder occurs during turnover and their simulation-based argument about the ND1 loop is contradicted in part by the disorder in their own structure.

Please explain the assumptions made, if any, in CAVER modelling. How does the CAVER modelling take into account uncertainty in side chain positions? There is weak to no density for many key side chains in this region, as seen in Fig S5 for the WT enzyme.

Experimental errors are needed for the activity of the purified complexes. (Extended Data Table 1)

Minor Comments:

Line 91: Please indicate the residues on the core subunits interacting with LYRM6 residues. Maybe add a figure or a table indicating major interactions.

Line 94: L42 of LYRM6 is close to C-terminus of the helix not in loop

Line 100: Both mutants Y43A and F89A seem to have similar effect although purified Y43A mutant complex I seems to have a little higher activity. What is the basis for choosing F89A mutant for further structural studies? Please discuss the observations, if any, made with Y43A and L42A mutants for comparison.

Line 119: It is quite remarkable indeed that a single substitution affects structural elements $\sim 50 \text{ \AA}$ away from the region of substitution. Is the missing density on NDUFA9 a consequence of poor local resolution due to some other factors or is it due to the mutation on LYRM6? The authors should briefly discuss the effect of the mutation on other accessory subunits in the vicinity of LYRM6. Please elaborate the tilting of the peripheral arm. How does the tilting of the peripheral arm affect other accessory subunits?

Comment on interactions with lipids.

Line 126: Could the authors elaborate on the structural differences between WT and F89A mutant that are comparable to A/D transition?

Line 175: Please provide CAVER based structural analysis of F89A LYRM6 complex I and compare the substrate protonation channel between WT and the F89A mutant

Line 180: K130 ND1 is $\sim 11 \text{ \AA}$ from E39 ND3 in the *Yarrowia* complex I structure. The conformation from the wild type structure should be shown to indicate the degree of motion in this loop during the wild type simulation.

Reviewer #2 (Remarks to the Author):

The authors elucidate the role of accessory subunit LYRM6 (NDUFA6) in mitochondrial respiratory complex I. Based on a new cryo-EM structure of complex I of a F89A(LYRM6) mutant and MD simulations they suggest that the interaction of LYRM6 with TMH1-2 loop in subunit ND3 maintains an open proton pathway to the central Q redox site.

One of the major results is that a tunnel leading from the N side to the Q redox site is open in WT and in a constitutively active mutant (Q133C) but partially closed in the inactive F89A mutant. The data are interpreted to mean that this tunnel allows protons to enter the Q redox site, as required by the ubiquinone reduction, and that in the

inactive mutant, protons cannot diffuse to the Q site.

I will primarily comment on the MD simulation aspects of the paper.

MAJOR COMMENTS

(1) There is no direct evidence for proton movement given (which is hard). The authors rely on the assumption that water accessibility is a good proxy for the proton movement through the enzyme in order to reach their conclusion regarding the functional relevance of the tunnel.

Ideally, simulations for proton transfer in the aqueous phase would be performed but I recognize that this would be beyond the scope of this paper. However, the following needs to be considered:

- Their assumption needs to be justified and explained.
- How does a "water occupancy of 0.2" (Fig 3) relate to the ability of protons to move through the same region? For instance, the WT hydration path (contour 20%) appears already broken near V212.
- For Grotthuis-type mechanisms, a water wire with correct water orientations is necessary (see e.g., the literature on aquaporins and why they do not conduct H⁺ or on H⁺ pumping in bacteriorhodopsin). Are water wires present?
- Electrostatics may also play a role if the proton is believed to mostly travel as a solvated proton; in this case a simple Poisson-Boltzmann "Born" energy calculation could be performed for an approximate potential of mean force.

(2) The MD simulations of F98A appear to approach WT levels for $t > 900$ ns (E39 - V212 distance), Fig 3i. It looks as if for longer times there's little difference and the convergence of the results is not clear.

- Extend simulations to show convergence.
- Analyze water density for parts of the simulations when WT and mutant are similar/different (roughly 0-200ns, 200ns-900ns, >900ns). Is the density from the last part of the mutants more "open" again and more similar to WT?

(3) A possible computational test for the hypothesis that F89A(LYRM6) is responsible for the closure of the (putative) proton pathway, the authors could start with the WT structure, mutate F89A(LYRM6) and see if the TMH1-2(ND3) loop relaxes during MD simulations. In this way, the result would not be influenced by the cryo-EM structure.

(This is a suggestion that would considerably strengthen the

authors' argument but I recognize that it is not clear how long it will take for the structure to relax.)

(4) (line 210) "This indicates that amino acid residues from the accessory subunit provide indispensable stabilizing interactions for efficient functioning of mitochondrial complex I" – the structural and simulation data really only refer to F89A and these results are now generalized in this statement. The functional data also suggest stabilizing roles for L42 and Y43 (Y43A showed the same low activity in mitochondrial membranes as F89A, see Ext Data Table 1).

The authors' argument would be strengthened and made more robust if at least MD simulations of Y43A showed similar results as the F89A simulations.

METHODS

(5) What was the reason for simulating all residues with standard protonation states (except His95(S2) and changes to His91)? Were any pKa calculations performed to justify the protonation states?

(6) Only subsystem of the (very big!) complex I was simulated, which raises the question if this setup introduces any artifacts, especially as the authors point out the presence of very long range allosteric interactions in complex I.

- The authors should show any evidence that they have that this approach does not disturb the structure and dynamics near the Q site.

Alternatively, at least clearly state that the simulations are performed under the unproven assumption that they properly model their region of interest.

- Show an image of simulation system.

- Show the structural stability, in particular CA RMSD timeseries for the simulations, secondary structure, and RMSF so that readers can form a better picture of the quality of the simulations.

(7) Add time constants for thermostats and barostats and the details for the non-bonded interactions (PME?, LJ cut-offs).

(8) Please state versions of software used (PyMOL, GROMACS, VMD, CAVER).

(For instance, recent versions of Gromacs had a bug that lead to unstable simulations of membrane proteins with the CHARMM force field.)

(9) Ref 68 is not the appropriate citation for Gromacs ... look at the output log file for proper references.

(10) Explain how the 3D water "occupancy" map (gray mesh in Fig 3b,c) was computed. The legend to Ext Data Table 3 indicates that only waters within 4 Å of certain residues were selected.

- Define "occupancy" and explain why it was used instead of the physical density.

- Was the occupancy calculated with VMD's VolMap plugin or by some other means? Was it calculated from point-particles or atomic spheres?

- Why did the authors sub-select water molecules instead of just computing the occupancy/density from all waters?

- Around the CAVER-tunnel region, is there a difference between the occupancy from the subselection vs. all waters?

MINOR COMMENTS

(11) State that *Yarrowia lipolytica* is a yeast -- not every reader might know this without looking it up... Also state the organism when talking about the F89A cryo-EM structure for the first time; it might be useful to mention that this is a structure from a eukaryote.

(12) Fig 3de (and Ext Data Fig 8) - why are the distance distributions scaled to 1? They should be normalized so that the area under the distribution equals 1 so that they F89A and WT can be compared directly.

Furthermore, it is confusing to label the y-axis of the histograms as "occupancy" when at the same time the water density is also called occupancy.

(13) Clarify in the text that there is a disulfide bond C40(ND3)-C133(S7) present in the Q133C mutant simulation.

Reviewer #3 (Remarks to the Author):

This work addresses key structure-function relationships in complex I of the mitochondrial respiratory chain. Perhaps surprisingly, a key role of one of the nuclear encoded "additional" subunits in complex I activity is revealed by careful and well-controlled experimentation and site-directed mutagenesis. The

3D structure of one of the site-directed mutant proteins suggests that the deficiency may be the result of blocking the pathway for protonation of the reduced ubiquinone in the active site. MD simulations support this assessment.

I think this is very high quality work that should be published, and I only have two general comments & requests of possible amendment.

First, can the authors suggest how the bacterial complex I counterpart (which lacks the nuclear subunit) does the "job", and/or whether the nuclear subunit might provide a control mechanism?

Secondly, the figures provided suggested to me that the protonation pathway (of the "chemical" proton) ended up perhaps half-way of the length of the ubiquinone channel, i.e. quite far from the site where the bound Q would have to be to receive the electron(s) from the nearest Fe/S center. As far as I could see, there is no comment on this in the paper. I mean, why isn't protonation of Q "allowed" in the position where ubiquinone receives the electron(s)?

REVIEWER COMMENTS

We thank all the reviewers for evaluation of the manuscript and their valuable suggestions.

Reviewer #1 (Remarks to the Author):

The manuscript “Essential role of accessory subunit LYRM6 in the mechanism of mitochondrial complex I” by E.G. Yoga and colleagues discusses the role of accessory subunit LYRM6 in the complex I redox pathway. The authors have determined the electron cryo-microscopy structure of F89A LYRM6 mutant of complex I at 3.0 Å and also performed fully atomistic MD simulations of WT and F89A LYRM6 mutant complex I. Their analysis of the structure reveals how a single substitution F89A in LYRM6 results in long-range structural changes that affect the loop cluster formed by core subunits ND1, ND3 and NDUFS2. In conclusion the paper discusses how LYRM6 stabilizes the loop cluster that may act as a single structural unit to control the flow of protons required in Q redox chemistry through a newly discovered proton access pathway.

The manuscript is well written and clear.

Major comments:

The authors have previously shown that LYRM6 is essential for complex I activity (PMID: 24706851). The current manuscript “Essential role of accessory subunit LYRM6 in the mechanism of mitochondrial complex I” does not provide significant further insight. The observations made are mostly descriptive and conclusions drawn lack thorough experimental evidence. The observations lead the authors to make predictions about a specific mechanism linking LYRM6 to complex I turnover but these predictions are not tested.

In our previous work, we showed that complex I is inactive in the absence of LYRM6. However, the reason for the complete loss of activity had remained unclear. Here we investigate the interface of LYRM6 with central subunits and explain in molecular details the impact of an accessory subunit on complex I function. Our combination of site-directed mutagenesis, high-resolution structural information and molecular simulations offered clues on an important element in the catalytic mechanism of complex I. In the revised version of our manuscript, we provide further evidence for the proposed proton access pathway to the Q reduction site, also by including new multiscale simulations. We also performed site-directed mutagenesis of tunnel residues from central subunits NDUFS2 and NDUFS7 that were identified by CAVER analysis and MD simulations. Eight of ten tunnel mutants showed loss of complex I activity indicating importance of the new tunnel for complex I function (updated Supplementary Table 3).

Also key observations that have been previously made from the authors own work are omitted from the discussion, these observations significantly impact on the interpretation of the results presented here. For instance, the authors have previously solved the structure of *Y. lipolytica* complex I during turnover in which they show disorder in the ND3 loop and many of the other loops and regions discussed here (eLife 2018;7:e39213, not referenced in this current manuscript). The flexible ND3 loop in the current structure therefore is not sufficient to explain reduced complex I activity. The two claims of “disorder means not functional” and “disorder during normal turnover” made in this and the previous work appear to be at odds and this contradiction needs to be addressed.

The structure of complex I under turnover (pdb file 6GCS) was determined at a lower resolution of 4.5 Å and compared with a structure of complex I in the deactive state at a resolution of 4.3 Å [Parey et al.,

2018, eLife]. In both structures, the central part of the ND3 loop was not modelled because of insufficient density. In that study, we observed that the ND3 loop appears to be flexible (disordered) in both states. It was indeed shown by Cabrera-Orefice et al. [2018, Nat. Commun., cited and discussed in the manuscript] that rearrangement of the ND3 loop is essential for coupling electron transfer with proton pumping, thus, coordinated conformational changes in the loop (“not disorder”) must occur during turnover. In our current work, we shed light on these functionally relevant conformational changes by structural biology and molecular simulation approaches.

We here compare our new 3.0 Å structure of the F89A-LYRM6 mutant with our recently published 3.2 Å structure of WT complex I (PDB 6RFR, deactive state). In the 3.2 Å high-resolution WT structure, the ND3 loop is resolved and modeled, while density for a substantial part of the loop is clearly missing in the F89A^{LYRM6} mutant (at 3.0 Å). Therefore, we can here state that the loop is relatively more mobile (disordered) in mutant and/or accesses conformational states attributed with the loss of function. Furthermore, we can identify the important contacts with LYRM6 that hold the loop in place (position 42-44 of the ND3 loop) and conclude that accessory subunit LYRM6 stabilizes conformations of the ND3 loop. Considering the work by Cabrera-Orefice et al. and our comparison of several complex I structures, we know that movement of the ND3 loop segment comprising the strictly conserved EXG motif (position 39-41 of the ND3 loop) is critical for complex I function. Here, we provide molecular insights into this critical region, and conclude that in the absence of stabilizing interactions with LYRM6 at position 42-44 of the ND3 loop, the controlled movement of the EXG motif of the ND3 loop is no longer possible and activity is lost. Therefore, we agree with the referee’s point that flexibility/disorder alone does not account for mutant inactivity, instead, mutation perturbs strictly regulated movements of ND3 loop. This point is now made clearer in the text and supplementary material to avoid confusion.

The authors argue that LYRM6 is necessary to hold the loop cluster formed by the core subunits in place to enable transport of protons required for Q redox chemistry. However, they do not discuss, what holds the loop cluster in place in bacterial enzyme where a homologue of LYRM6 is absent? The authors should consider carrying out fully atomistic MD simulations of the bacterial enzyme, which lacks LYRM6, and comparing it to their current simulations.

We have now performed long timescale MD simulations (two independent simulation replica, see updated Supplementary Table 4) of *Thermus* enzyme, and studied the dynamics of the central loop cluster. Primarily, we find that the ND3 loop is much more mobile in *Thermus* enzyme compared to *Yarrowia*, because it is more exposed to the solvent in the former due to the missing LYRM6 subunit (new Supplementary Fig. 13). We also performed simulations of a larger atomic model of WT *Yarrowia* complex I comprising ca. 1.3 million atoms (updated Supplementary Table 4) and find ND3 loop to be less mobile, similar to our small-sized models (new Supplementary Fig. 13). We have included these data and discussion in the revised manuscript, and it clearly suggests that accessory subunit LYRM6 limits the conformational flexibility of ND3 loop and allows its more coordinated movements. We conclude that in *Yarrowia* enzyme presence of accessory subunit LYRM6, which limits conformational flexibility of ND3 loop, may provide an additional layer of mechanistic control of complex I function, such as more precise gating of substrate proton transfer in comparison to bacterial enzyme.

Interestingly, a similar picture emerges from extensively studied cytochrome *c* oxidase, where a 2 or 3 subunit bacterial enzyme has been found to have lower efficiency (specially under high pmf conditions) compared to mitochondrial version that has additional gating mechanisms and accessory subunits [Rauhamäki and Wikström, BBA Bioenergetics, 2014]. Mitochondria are complex organelles for robust

and powerful energy production. Respiratory complexes of eukaryotes are a) more complex (because of accessory subunits), b) mainly organized in supercomplexes, c) work under high pmf, d) must be controlled according to cell/organism requirements, and e) have to adapt to different conditions (e.g. hypoxia). Our study suggests how complex I activity might have evolved to higher efficiency by addition of accessory subunit LYRM6.

Insufficient justification is given for the choice of residues mutated in this study. If they are interested in conserved features of complex I regulation by LYRM6, why did they not target conserved residues? It is unclear why this poorly conserved set of residues were targeted. If they were structure based what were the specific criteria used?

The selection of LYRM6 residues was based on structure. We observed that two loops of accessory LYRM6 contact highly conserved or/and important structural units at the interface of the matrix and membrane arm of complex I. We decided to perform site-directed mutagenesis in the accessory loop cluster formed by LYRM6 that contacts central subunits. After superimposing different eukaryotic complex I structures from yeast and mammals we observed that the folding and position of LYRM6 is overall very similar (updated Supplementary Fig. 1). We agree that residues in the proximal loops of LYRM6 are not highly conserved from yeast to mammals, however in all eukaryotes these loops interact with the core subunits.

The authors claim that F89A, Y43A substitution reduces complex I activity via interaction with the ND3 loop residues. However, they do not show any details of the specific interactions with between LYRM6 and the ND3 loop. If they want to demonstrate the importance of these interactions, they should consider carrying out complimentary mutations substituting the residues interacting with F89 and Y43 in the loop cluster and studying the effect of those substitutions on complex I activity. In other words, the authors need to make a prediction and test that prediction to prove their hypothesis, otherwise the study is merely descriptive and we already know LYRM6 is important for activity.

The authors should discuss how the Y43A mutant would result in decreased activity of the complex. Y43 is positioned to avoid any steric clashes with the loop.

L42, Y43 and F89 of accessory subunit LYRM6 form a loop arrangement connecting the two adjacent loops of LYRM6. F89^{LYRM6} protrudes most prominently towards the ND3 loop. We inserted an additional panel in updated Supplementary Fig. 2 to show the arrangement of these residues and we provide additional MD simulations of Y43A-LYRM6 in Supplementary Fig. 12. We have indeed thought about complimentary exchanges in the ND3 loop. However, please note that central subunit ND3 of mitochondrial complex I is encoded by mitochondrial DNA (mtDNA). For eukaryotic complex I, no experimental system is available to generate point mutations in mtDNA encoded complex I subunits at present and this is the case for all mitochondrial systems. We showed pathogenic human mutations located in the TMH1-2^{ND3} loop in Supplementary Fig. 6 (new Supplementary Fig. 8) to provide more insights about the critical segment of the ND3 loop. We now modelled the Y43A mutant and performed new MD simulations (updated Supplementary Table 4). We find similar structural changes as observed in F89A mutant (shorter T43^{ND3}-F89^{LYRM6} and longer C40^{ND3}-H91^{S2} distances), which also provides reasoning for the decreased activity of Y43A mutant.

The ND1 loop that “blocks” the hydration path in the simulation of the mutant is disordered in the structure indicating conformational flexibility. Flexibility calls into question the degree to which the

mutant stabilizes this "blocking" conformation of the loop. How do the authors reconcile these contradictory results?

We extended our WT and F89A-LYRM6 simulations to 2.5 μ s per simulation replica (see updated Supplementary Table 4). Based on this extensive simulation sampling, we find that the ND3 loop segment is indeed more mobile in F89A-LYRM6 case and samples conformational states that are less occupied in WT (updated Fig. 3 and new Supplementary Fig. 12). Similarly, the ND1 loop stabilizes the "blocking" conformation more in mutant as compared to WT (see revised Fig. 3). We think that the conformational flexibility, as seen in F89A cryo-EM experiment, can also be interpreted as additional conformational states visited by ND3 and ND1 loops, which are less populated in WT enzyme. The MD observed states are associated with the loss of activity in mutant cases.

This is also substantiated by our further analysis (as a suggestion from another referee) and shows a clear correspondence between "open/closed" transitions of TMH5-6 hydrophobic gate (Val212^{ND1}-Glu39^{ND3} distance) and presence/absence of hydration in the region (see Fig. 3 and Supplementary Figs 15 and 16). For instance, out of the total three F89A-LYRM6 simulation replicas, two show "closed" V-E gate, whereas the third (Fig. 3 and Supplementary Fig. 16) show it to be "open", but at around 2 μ s, the V-E distance starts to relax to the "closed" state. Similarly, two WT replicas reveal "open" V-E conformation, whereas the third one is closed, but distance also starts to increase in the last part of the simulation (Fig. 3 and Supplementary Fig. 15). These data suggest that both "open/closed" events occur in WT and mutant. However, there is preference for the channel to be open in WT case (Fig. 3). This proposed gating mechanism is probably part of the catalytic cycle and is present in both bacterial and mitochondrial enzymes. This is now discussed in the text.

The authors don't show convincingly the mechanism by which this mutant is disrupting activity. According to their own work, disorder occurs during turnover and their simulation-based argument about the ND1 loop is contradicted in part by the disorder in their own structure.

In our view, the generalized statement that turnover causes "disorder" is a misleading oversimplification. We have proposed concerted rearrangement of loops, i.e. controlled movement, but no "disorder". In our comprehensive approach, we are able to compare results from MD simulations and structure determination. If density in a cryo-EM map is missing, several reasons can be discussed. It could be disorder in the sense of complete flexibility but it could also mean the population of several defined substates. This information cannot be extracted from the map. Nevertheless, in case of the ND1 loop discussed here, it is interesting to note that at least a difference between WT and mutant structures is observable. The suggestion that the ND1 loop (TMH5-6 hydrophobic segment) blocks the proton access path is based on our MD simulations, in part also supported by structural data (Supplementary Fig. 14). It is true that the loop segment was not modelled in the mutant structure and hence we have no structural evidence for the blocking conformation. However, in our view this is not a contradiction because the simulations support the existence of differently populated substates that might ultimately preclude observation of sufficiently strong density for model building. We have made this point more clear in the discussion.

Please explain the assumptions made, if any, in CAVER modelling. How does the CAVER modelling take into account uncertainty in side chain positions? There is weak to no density for many key side chains in this region, as seen in Fig S5 for the WT enzyme.

We thank the reviewer for this valuable comment. Cryo-EM electron maps suffer from loss of structural information from carboxylate groups of most acidic sidechains of Glu and Asp residues induced by radiation damage [Vonck and Mills, *Curr. Opin. Struct. Biol.*, 2017]. In many cases the side chains of these residues are not resolved in cryo-EM maps. However, the backbones of the loops are clearly visible and modelling of Glu sidechains follows sterical requirements. A complete PDB protein model of WT enzyme was used for Caver analysis.

Experimental errors are needed for the activity of the purified complexes. (Extended Data Table 1)
We added the required information to updated Supplementary Table 1.

Minor Comments:

Line 91: Please indicate the residues on the core subunits interacting with LYRM6 residues. Maybe add a figure or a table indicating major interactions.

We added a panel in updated Supplementary Fig. 2.

Line 94: L42 of LYRM6 is close to C-terminus of the helix not in loop

In some structures of mitochondrial complex I, the C-terminal end of helix $\alpha 1^{\text{LYRM6}}$ was modelled as loop. To clarify the situation in updated Supplementary Fig. 1a, this part of *Y. lipolytica* LYRM6 sequence has been labelled $\alpha 1'$.

Line 100: Both mutants Y43A and F89A seem to have similar effect although purified Y43A mutant complex I seems to have a little higher activity. What is the basis for choosing F89A mutant for further structural studies? Please discuss the observations, if any, made with Y43A and L42A mutants for comparison.

Indeed, L42, Y43 and F89 of accessory LYRM6 form a loop arrangement. In fact, F89^{LYRM6} directly interacts with the ND3 loop and results from site-directed mutagenesis of F89^{LYRM6} showed dramatic loss of complex I function in F89A^{LYRM6} mutant. Analysis of WT structure suggested that the accessory loop arrangement formed by L42, Y43 and F89 keeps F89^{LYRM6} in place. Thus, we were interested in the cryo-EM structure of F89A^{LYRM6} mutant to gain information about the most critical interaction point between accessory LYRM6 and ND3 loop.

In the main text we added:

“Residue F89^{LYRM6} of the accessory LYRM6 loop forms a prominent supporting structural element for the TMH1-2^{ND3} loop (Supplementary Fig. 2).”

Line 119: It is quite remarkable indeed that a single substitution affects structural elements ~ 50 Å away from the region of substitution. Is the missing density on NDUFA9 a consequence of poor local resolution due to some other factors or is it due to the mutation on LYRM6? The authors should briefly discuss the effect of the mutation on other accessory subunits in the vicinity of LYRM6. Please elaborate the tilting of the peripheral arm. How does the tilting of the peripheral arm affect other accessory subunits? Comment on interactions with lipids.

We have shown that the C-terminal domain of peripheral arm subunit NDUFA9 binds three lipid molecules [Parey et al., *Sci. Adv.*, 2019]. These lipids might be important to anchor NDUFA9 to the membrane and for stabilizing the L-shaped arrangement of peripheral and membrane arm. In this study we observed disorder in the lipid binding domain of NDUFA9 in the F89A mutant and tilting of the whole peripheral arm. At this stage it remains unclear whether disturbed lipid binding is cause or consequence of the structural perturbation. We added new Supplementary Fig. 6 and the legend comments on the tilting of the matrix arm and on the interactions of NDUFA9 with lipids.

Line 126: Could the authors elaborate on the structural differences between WT and F89A mutant that are comparable to A/D transition?

Disorder in mutant F89A^{LYRM6} and in mouse D state are very similar, but in mouse the A/D transition is reversible by turnover. We added new Supplementary Fig. 7 to show similarity of these two states.

Line 175: Please provide CAVER based structural analysis of F89A LYRM6 complex I and compare the substrate protonation channel between WT and the F89A mutant

We show CAVER analyses of WT and F89A^{LYRM6} mutant in new Supplementary Fig. 10 to show the tunnel residues of our updated mutagenesis study and to show disruption of the CAVER tunnel in the mutant.

Line 180: K130 ND1 is ~11Å from E39 ND3 in the Yarrowia complex I structure. The conformation from the wild type structure should be shown to indicate the degree of motion in this loop during the wild type simulation.

This is now shown in updated Fig. 3.

Reviewer #2 (Remarks to the Author):

The authors elucidate the role of accessory subunit LYRM6 (NDUFA6) in mitochondrial respiratory complex I. Based on a new cryo-EM structure of complex I of a F89A(LYRM6) mutant and MD simulations they suggest that the interaction of LYRM6 with TMH1-2 loop in subunit ND3 maintains an open proton pathway to the central Q redox site.

One of the major results is that a tunnel leading from the N side to the Q redox site is open in WT and in a constitutively active mutant (Q133C) but partially closed in the inactive F89A mutant. The data are interpreted to mean that this tunnel allows protons to enter the Q redox site, as required by the ubiquinone reduction, and that in the inactive mutant, protons cannot diffuse to the Q site.

I will primarily comment on the MD simulation aspects of the paper.

MAJOR COMMENTS

(1) There is no direct evidence for proton movement given (which is hard). The authors rely on the assumption that water accessibility is a good proxy for the proton movement through the enzyme in order to reach their conclusion regarding the functional relevance of the tunnel.

Ideally, simulations for proton transfer in the aqueous phase would be performed but I recognize that this would be beyond the

scope of this paper. However, the following needs to be considered:

- Their assumption needs to be justified and explained.

We have now performed hybrid QM/MM MD simulations on simulation snapshots selected from 'classically' hydrated protein to study explicit proton transfer reactions near the E39^{ND3} region of the proton channel (see new Supplementary Table 6). We find a hydronium (H₃O⁺) modeled near putative proton uptake site E39^{ND3} rapidly protonates it in two different QM/MM simulation setups, modeled with and without the observed K130^{ND1}-E39^{ND3} ion pair (see new Supplementary Fig. 11). In contrast, in QM/MM runs of F89A-mutant we did not see protonation of E39^{ND3} in these time scales. Furthermore, when protonation dynamics was studied between E39^{ND3} and H91^{S2}, proton rapidly stabilized on the latter, suggesting it can be transported towards the Q reduction site. With these simulations, we now provide a direct evidence that proton uptake can occur via this channel involving critical residue from ND3 subunit.

- How does a "water occupancy of 0.2" (Fig 3) relate to the ability of protons to move through the same region? For instance, the WT hydration path (contour 20%) appears already broken near V212.

The water occupancy value is somewhat arbitrary. We have chosen the value to display water map and protein in most clear fashion possible. Also, the same value of occupancy is chosen for all our analysis (either mutant or WT simulations). Water occupancy maps in panels b and c in updated Fig. 3 are now replaced with new ones to show hydration more clearly.

- For Grotthuis-type mechanisms, a water wire with correct water orientations is necessary (see e.g., the literature on aquaporins and why they do not conduct H⁺ or on H⁺ pumping in bacteriorhodopsin). Are water wires present?

Yes, water wires are present. Please see the new Supplementary Fig. 11. Due to excessive hydration around putative proton uptake site E39^{ND3}, multiple water wires are present. The waters are oriented with their hydrogen-bonds pointing towards E39^{ND3}, which would sustain proton transport. These water wires indeed support proton transfer via Grotthus-like mechanism, as shown by our QM/MM molecular dynamics simulations (Supplementary Fig. 11).

- Electrostatics may also play a role if the proton is believed to mostly travel as a solvated proton; in this case a simple Poisson-Boltzmann "Born" energy calculation could be performed for an approximate potential of mean force.

Yes, electrostatics likely plays an important role in this proton transfer, which occurs via Grotthuss-like proton transfer through water molecules. We have now shown this by a more accurate treatment, QM/MM-based molecular dynamics, in which a positively charged hydronium (proton) gets attracted to the anionic E39^{ND3} (putative proton uptake site) and protonates it in two independent setups.

(2) The MD simulations of F98A appear to approach WT levels for $t > 900$ ns (E39 - V212 distance), Fig 3i. It looks as if for longer times there's little difference and the convergence of the results is not clear.

- Extend simulations to show convergence.

We have now extended the WT and F89A mutant simulations to 2.5 μ s, and also performed additional simulations (see updated Supplementary Table 4). Based on new data, we agree with the referee that conformational changes in very dynamic loop segments are somewhat slow to relax. For instance, out of the total three F89A simulation replicas, two show hydrophobic gate (based on V212^{ND1}-E39^{ND3} distance) to be overall "closed", whereas the third (updated Fig. 3) show it to be "open", but at around 2 μ s the V-E distance starts to relax to the "closed" state. Similarly, two WT replicas reveal "open" V-E conformation, whereas the third one is closed, but distance also starts to increase in the last part of the simulation. These data suggest that both "open/closed" events occur in WT and mutant. However, there is preference for channel to be open in WT case (updated Fig. 3). The quantitative aspects of this concerted loop dynamics will be studied in our future work with free energy simulations.

We also emphasize that additional simulation sampling clearly improved the statistics on T-F/A distance metric (Figs. 3 and Supplementary Fig. 12), which now clearly shows that in mutant, the selected segment is more mobile, in agreement with the cryo-EM data, and shows conformation different from WT.

- Analyze water density for parts of the simulations when WT and mutant are similar/different (roughly 0-200ns, 200ns-900ns, >900ns). Is the density from the last part of the mutants more "open" again and more similar to WT?

We thank the referee for this very valuable suggestion. We have now analyzed and compared water occupancy maps for different parts of the simulation in different trajectories of WT and mutant simulations (Supplementary Figs. 15 and 16) and this further consolidates our conclusions. We do find a clear correspondence between "open/closed" transitions of TMH5-6 hydrophobic gate (Val212^{ND1}-Glu39^{ND3} distance) and presence/absence of hydration in the region. This proposed gating mechanism is probably part of catalytic cycle and is present in both bacterial and mitochondrial enzymes. This is now discussed in the text.

(3) A possible computational test for the hypothesis that F89A(LYRM6) is responsible for the closure of the (putative) proton pathway, the authors could start with the WT structure, mutate F89A(LYRM6) and see if the TMH1-2(ND3) loop relaxes during MD simulations. In this way, the result would not be influenced by the cryo-EM structure.

(This is a suggestion that would considerably strengthen the authors' argument but I recognize that it is not clear how long it will take for the structure to relax.)

We have indeed performed all our simulations (WT and mutant) on WT structure. This is because missing cryo-EM density in mutant structure would not allow unambiguous modelling of backbone and sidechains. This has now been clarified in the revised methods section. And, starting from a WT conformation, we do see clear changes in the ND3 loop conformation and its higher mobility (revised Fig. 3 and Supplementary Fig. 12) in agreement with mutant cryo-EM data.

(4) (line 210) "This indicates that amino acid residues from the accessory subunit provide indispensable stabilizing interactions for efficient functioning of mitochondrial complex I" – the structural and simulation data really only refer to F89A and these results are now generalized in this statement. The functional data also suggest stabilizing roles for L42 and Y43 (Y43A showed the same low activity in mitochondrial membranes as F89A, see Ext Data Table 1).

The authors' argument would be strengthened and made more robust if at least MD simulations of Y43A showed similar results as the F89A simulations.

We have now modelled Y43A mutant and performed new MD simulations (multiple replicas, see updated Supplementary Table 4). We find, structural changes, similar to F89A simulations, are seen in Y43A case (lower T43^{ND3}-F89^{LYRM6} and higher C40^{ND3}-H91^{S2} distance, see Supplementary Fig. 12). In Y43A, ND3 loop assumes different conformations than WT, as in F89A, and is likely the reason for its inactivity.

METHODS

(5) What was the reason for simulating all residues with standard protonation states (except His95(S2) and changes to His91)? Were any pKa calculations performed to justify the protonation states?

Complex I consists of a very large number of titratable residues, most of which are present in the functionally critical regions of the enzyme (Q tunnel, E channel and antiporter-like subunits). Due to very large number of titratable sites and highly complex environment (cluster of charged residues present nearby, on dynamic loops, etc), we have avoided empirical or PB based pKa calculations, which have

limited accuracy. Furthermore, empirical pKa predictions show pKa to be close to 7 for many residues in complex I, which makes it further hard to select a particular protonation state and simulate. Therefore, we have relied on modeling specific states of residues in concern based on available structural and experimental data. In the current case, H95^{S2} forms an ion pair with the conserved D196^{S2} in *Yarrowia* and *Thermus* enzyme structures. This interaction has been studied extensively in our earlier works [Sharma et al., PNAS, 2015; Haapanen et al., 2019, Front. Chem.] and is found to be important for Q binding and its redox reactions. Experimental data also supports the importance of H95^{S2} [Grgic, J. Biol. Chem., 2004]. H91^{S2} from the same β 1- β 2 loop also forms ion pairs with E39^{ND3} in *Thermus* structure (see new Supplementary Fig. 9). Moreover, we have observed its elevated pKa in our earlier work [Warnau, Sharma et al., PNAS, 2018], thus we decided to simulate it in its two protonation states (neutral and cationic). See Supplementary Table 4.

(6) Only subsystem of the (very big!) complex I was simulated, which raises the question if this setup introduces any artifacts, especially as the authors point out the presence of very long range allosteric interactions in complex I.

- The authors should show any evidence that they have that this approach does not disturb the structure and dynamics near the Q site.

We have now simulated a larger model system of *Yarrowia* complex I (ca. 1.3 million atom model) and compared it to our smaller sized systems (ca. 452000 atoms). We find our smaller models to be stable. See response below and also the RMSD and secondary structure plots (Supplementary Figs. 18 and 19).

Alternatively, at least clearly state that the simulations are performed under the unproven assumption that they properly model their region of interest.

We have also added a note on it. Please see revised MD simulation methods section.

- Show an image of simulation system.

A figure has been added (see Supplementary Fig. 18).

- Show the structural stability, in particular CA RMSD timeseries

for the simulations, secondary structure, and RMSF so that readers

can form a better picture of the quality of the simulations.

We have now shown RMSD and secondary structure plots to show stability of model systems (see Supplementary Figs. 18 and 19). RMSF of selected segment is in Supplementary Fig. 13. Comparison between small and big model systems show that the truncation has a minor effect and small-scale systems are stable.

(7) Add time constants for thermostats and barostats and the details for the non-bonded interactions (PME?, LJ cut-offs).

These have now been added.

(8) Please state versions of software used (PyMOL, GROMACS, VMD, CAVER).

(For instance, recent versions of Gromacs had a bug that lead to unstable simulations of membrane proteins with the CHARMM force field.)

The version numbers have been added.

The Gromacs issue concerned the GPU-versions of Gromacs 2018 and possibly prior (see: <https://redmine.gromacs.org/issues/2845>). The fix was reported in release notes of 2018.8 (<http://manual.gromacs.org/documentation/2018.8/release-notes/2018/2018.8.html>). Later, in 2019 GPU-versions of Gromacs had some problems (<https://redmine.gromacs.org/issues/2867>). Again, a fix was introduced in 2019.4 release notes (<http://manual.gromacs.org/documentation/2019.4/release-notes/2019/2019.4.html>). The only GPU-version of Gromacs we have used is 2020.2. So far to our knowledge, CHARMM related errors have not been reported for this version. Moreover, the model system setups and production MD simulations in this work have been carefully performed and no instabilities are observed.

(9) Ref 68 is not the appropriate citation for Gromacs ... look at the output log file for proper references.

Reference is now fixed. Due to limitations imposed by the journal on total number of references, only the most relevant ones might be included.

(10) Explain how the 3D water "occupancy" map (gray mesh in Fig 3b,c) was computed. The legend to Ext Data Table 3 indicates that only waters within 4 Å of certain residues were selected.

- Define "occupancy" and explain why it was used instead of the physical density.

Occupancy was used because of its conceptually simpler definition, as described in VMD Volmap plugin (<https://www.ks.uiuc.edu/Research/vmd/vmd-1.9.1/ug/node153.html>).

- Was the occupancy calculated with VMD's VolMap plugin or by some other means? Was it calculated from point-particles or atomic spheres?

Yes, we have used VMD VolMap plugin. Atoms were treated as spheres.

- Why did the authors sub-select water molecules instead of just computing the occupancy/density from all waters?

The region surrounding the locus of interest is very hydrated and a clear figure for illustration purposes is very difficult to achieve with occupancy calculated from all waters in the region. Therefore, we selected water molecules around residues that would encompass the region from channel opening (E44^{LYRM6}/E39^{ND3}) to region close to Q tunnel (H91^{S2}). This is the proposed path for proton transfer. See Fig. 3a.

Most interestingly, some of the residues that were found close to the hydrated region were subjected to site directed mutagenesis (as a suggestion by another referee), and were found to be critical for complex I activity (see revised Supplementary Table 3). This further consolidates our conclusions.

- Around the CAVER-tunnel region, is there a difference between the

occupancy from the subselection vs. all waters?

We have visually checked on the graphics terminal and there are no differences in occupancy either done by subselection or all waters.

MINOR COMMENTS

(11) State that *Yarrowia lipolytica* is a yeast -- not every reader might know this without looking it up... Also state the organism when talking about the F89A cryo-EM structure for the first time; it might be useful to mention that this is a structure from a eukaryote.

This has now been mentioned already in the abstract. It has also been clarified at additional places in the revised text.

(12) Fig 3de (and Ext Data Fig 8) - why are the distance distributions scaled to 1? They should be normalized so that the area under the distribution equals 1 so that they F89A and WT can be compared directly.

We have now replotted all plots (updated simulation trajectory data) by normalizing to area under curve equal to 1.

Furthermore, it is confusing to label the y-axis of the histograms as "occupancy" when at the same time the water density is also called occupancy.

We agree with the referee, and we have now changed it to normalized counts.

(13) Clarify in the text that there is a disulfide bond

C40(ND3)-C133(S7) present in the Q133C mutant simulation.

This is now further emphasized in the text.

Reviewer #3 (Remarks to the Author):

This work addresses key structure-function relationships in complex I of the mitochondrial respiratory chain. Perhaps surprisingly, a key role of one of the nuclear encoded "additional" subunits in complex I activity is revealed by careful and well-controlled experimentation and site-directed mutagenesis. The 3D structure of one of the site-directed mutant proteins suggests that the deficiency may be the result of blocking the pathway for protonation of the reduced ubiquinone in the active site. MD simulations support this assessment.

I think this is very high quality work that should be published, and I only have two general comments & requests of possible amendment.

First, can the authors suggest how the bacterial complex I counterpart (which lacks the nuclear subunit) does the "job", and/or whether the nuclear subunit might provide a control mechanism?

This is an interesting point and we concur with referee's point that accessory subunits may provide an additional control mechanism. We have now discussed it in the light of newly added simulation data on bacterial complex I (*Thermus thermophilus*, see Supplementary Table 4). It appears that in *Yarrowia lipolytica* enzyme accessory subunit LYRM6 provides a scaffolding to the functionally critical loop of core ND3 subunit, and thus restricts and guides its conformational movement. In case of *Thermus thermophilus* complex I, due to the lack of accessory LYRM6 subunit, the ND3 loop is much more mobile (Supplementary Fig. 13). We suggest that a more restricted and coordinated movement of ND3 loop optimizes the timing of substrate proton transfer to the Q redox site. While in *Thermus thermophilus* enzyme, we propose that the protonation of negative intermediates occurs either prematurely or in delayed fashion, and is thus not optimally controlled in comparison to eukaryotic enzyme.

Secondly, the figures provided suggested to me that the protonation pathway (of the "chemical" proton) ended up perhaps half-way of the length of the ubiquinone channel, i.e. quite far from the site where the bound Q would have to be to receive the electron(s) from the nearest Fe/S center. As far as I could see, there is no comment on this in the paper. I mean, why isn't protonation of Q "allowed" in the position where ubiquinone receives the electron(s)?

This is also a very interesting point and we have added a brief commentary on this in Supplementary Fig. 17. The exact identity of proton holes created upon two-electron reduction of Q is not known. It could be the anionic Q itself or amino acid residues left deprotonated after PCET reaction [Sharma et al., PNAS, 2015]. Protonation of anionic Q at a site different from the electron accepting site near N2 is in agreement with the two-state stabilization change mechanism [Brandt et al., BBA, 2011]. In second case, protonation of proton holes can occur via water molecules that diffuse into the channel only after QH₂ has departed [Wikström, Chem Rev, 2015; Parey et al., Sci Adv, 2019].

We updated Supplementary Fig. 17 (revised numbering) to introduce the model for the protonation site "half-way the length of Q channel".

In the main text we added:

“Uncontrolled protonation of reduced Q intermediates by bulk protons would result in heat production instead of controlled energy conversion for generation of proton pumping strokes.”

REVIEWERS' COMMENTS

Reviewer #1 (Remarks to the Author):

The manuscript "Essential role of accessory subunit LYRM6 in the mechanism of mitochondrial complex I" by E.G. Yoga and colleagues discusses the role of accessory subunit LYRM6 in the complex I catalytic cycle. The authors have determined the electron cryo-microscopy structure of F89ALYRM6 mutant of complex I at 3.0 Å and also performed fully atomistic MD simulations of WT, F89ALYRM6 and Y43ALYRM6 mutant complex I, as well as the bacterial complex I from *Thermus thermophilus* that lacks subunit LYRM6. Their analysis of the structure reveals how a single substitution F89A in LYRM6 may result in long-range structural changes that affect the loop cluster formed by core subunits ND1, ND3 and NDUFS2. They propose the existence of a proton pathway formed by these loops that is key to protonated quinone upon reduction of the substrate. They perform additional functional analysis as well as QM/MM molecular dynamics simulations to corroborate this possible proton pathway's existence and importance in the turnover of the complex. In conclusion, the paper discusses how LYRM6 stabilizes the loop cluster that may act as a single structural unit to control the flow of protons required in Q redox chemistry through this newly proposed proton access pathway.

The manuscript is well written and clear.

The additional analysis, mutagenesis and QM/MM simulations all act to strengthen the conclusions of the manuscript.

While this manuscript has been in development two major complex I structural papers have been released. One describing a higher resolution structure of *Y. lipolytica* complex I that reveals the positions of several water molecules (1Grba DN, Hirst J. Mitochondrial complex I structure reveals ordered water molecules for catalysis and proton translocation. *Nat Struct Mol Biol.* July 2020:1-20. doi:10.1038/s41594-020-0473-x). The other claiming to have described the coupling mechanism in mammalian mitochondrial complex I (Kampjut D, Sazanov LA. The coupling mechanism of mammalian respiratory complex I. *Science.* September 2020:eabc4209-eabc4218. doi:10.1126/science.abc4209). The authors should address these new developments in their manuscript with a focus on two important aspects that impact their findings:

1. Does the positions of the observed water molecules in the *Y. lipolytica* structure agree with those of the MD hydrated models? Are they consistent with the proposed protonation pathway defined by ND1, ND3 and NDUFS2?
2. Kampjut and Sazanov propose a different pathway for quinone protonation (via the core hydrophobic subunits ND4L Glu34 and Glu70), this is in direct conflict with the ND1, ND3 and NDUFS2 pathway being proposed here and needs to be discussed.

Minor comment:

The discussion on the bacterial simulations are quite speculative and should be rewritten. For example:

Lines 263-237.

"The data revealed a higher level of mobility in the TMH1-2ND3 loop in bacterial complex I compared to the eukaryotic enzyme because of the absence of LYRM6 accessory subunit"

It is unclear, given all the differences in sequence and subunits between the bacterial and *Y. lipolytica* structures, how the authors conclude that the differences they observe are solely due to the absence of LYRM6.

Lines 240-245.

"We propose that bacterial enzymes utilize the same proton transfer pathway, but due to lack of coordinated movement of ND3 loop, the protonation of anionic intermediates at the Q redox site is not optimally timed as in eukaryotic enzyme, instead may occur prematurely or in a delayed fashion. It is

known that mitochondrial proton-motive force (PMF) reaches values up to 200 mV and LYRM6-backed functional movements of THM1-2ND3 loop might support the enzyme to pump protons at high PMF.” Is there experimental data supporting the difference in timing between the bacterial and eukaryotic enzyme? If so please add the citation, if not please remove this discussion as it is too speculative.

Reviewer #2 (Remarks to the Author):

The authors addressed all my previous comments. They went beyond what I suggested, in particular they performed QM/MM calculations that present strong computational evidence for their hypothesis of proton transport through the channel and classical MD simulations of the whole complex I system that validate their approach to perform simulations on a smaller subsystem.

Reviewer #3 (Remarks to the Author):

The authors have satisfactorily responded to my two questions

Response to referee # 1. Our response in blue bold font.

1. Does the positions of the observed water molecules in the *Y. lipolytica* structure agree with those of the MD hydrated models? Are they consistent with the proposed protonation pathway defined by ND1, ND3 and NDUFS2?

Yes, comparison shows some similarities between the water positions observed in the recent *Y. lipolytica* complex I structure (Grba and Hirst 2020 Nature Str. Mol. Biol.) and our MD hydrated models. However, no water molecules are observed in the structure at the location of the proposed proton transfer pathway. This region is conformationally mobile and it is very likely that highly dynamic water molecules are not captured in the cryo EM experiment. However, some water molecules are observed in the high-resolution sheep complex I structures (Kampjut and Sazanov 2020 Science) in the region analyzed in our work, which would support our proposed proton transfer route. The detailed analysis of hydration and its functional meaning will be discussed in our future works.

2. Kampjut and Sazanov propose a different pathway for quinone protonation (via the core hydrophobic subunits ND4L Glu34 and Glu70), this is in direct conflict with the ND1, ND3 and NDUFS2 pathway being proposed here and needs to be discussed.

Indeed, the substrate proton transfer pathways proposed by Kampjut and Sazanov (and also by Grba and Hirst) are different from our proposal. Further work is needed to resolve these issues. Nevertheless, our proposal is supported by structural, biochemical and computational analysis, and we have briefly discussed it in the text.

Lines 263-237.

“The data revealed a higher level of mobility in the TMH1-2ND3 loop in bacterial complex I compared to the eukaryotic enzyme because of the absence of LYRM6 accessory subunit”;

It is unclear, given all the differences in sequence and subunits between the bacterial and *Y. lipolytica* structures, how the authors conclude that the differences they observe are solely due to the absence of LYRM6.

LYRM6 is the only accessory subunit that is the closest to the region of interest (ND3 loop segment) that shows maximal fluctuation in bacterial complex I simulations compared to *Y. lipolytica* complex I simulations (SI Fig. 13). The other accessory subunit NDUFA9/39-kDa is positioned further away. Thus, we can conclude that absence of LYRM6 can directly affect the dynamics of ND3 loop. However, we agree with referee’s point that there are sequence and subunit differences between the two enzymes, which may also affect this observed behaviour (e.g. long range effects). Therefore, we have revised the statement

accordingly.

Lines 240-245.

We propose that bacterial enzymes utilize the same proton transfer pathway, but due to lack of coordinated movement of ND3 loop, the protonation of anionic intermediates at the Q redox site is not optimally timed as in eukaryotic enzyme, instead may occur prematurely or in a delayed fashion. It is known that mitochondrial proton-motive force (PMF) reaches values up to 200 mV and LYRM6-backed functional movements of THM1-2ND3 loop might support the enzyme to pump protons at high PMF;

Is there experimental data supporting the difference in timing between the bacterial and eukaryotic enzyme? If so please add the citation, if not please remove this discussion as it is too speculative.

To the best of our knowledge, there is no such experimental data. Therefore, we have decided to amend the proposed viewpoint accordingly (see revised text).